# Study on Time-Dependent Failure Mechanisms and CBAG Differential Support Technology of Roadway in Steeply Inclined Coal Seam

**Zhengzheng Xie** [1], **Jin Wang** [1,*], **Nong Zhang** [1,2], **Feng Guo** [1], **Zhe He** [1], **Zhe Xiang** [1] and **Chenghao Zhang** [3]

1    State Key Laboratory of Coal Resources and Safe Mining, School of Mines,
     China University of Mining and Technology, Xuzhou 221116, China
2    School of Civil Engineering, Xuzhou University of Technology, Xuzhou 221018, China
3    Laboratory of Geotechnics, Department of Civil Engineering, Ghent University, Zwijnaarde, 9052 Gent, Belgium
*    Correspondence: wang_jin@cumt.edu.cn; Tel.: +86-182-0520-3728

**Abstract:** In Sichuan Province, China, most coal seams that are mined are steeply inclined; their roadways' surrounding rocks are asymmetric, with non-equilibrium deformations and unstable anchorage structures, thus making major safety hazards highly likely. Using field observations and a universal distinct element code (UDEC) numerical simulation method, this paper analyzed the time-dependent failure of the ventilation roadway of Working Face 1961 of the Zhaojiaba Mine, revealing the preconditions for such damage and a bidirectional deterioration mechanism for the deformation as well as stress of surrounding rocks. Moreover, this paper built an anchorage mechanical model for the thick layer of the roadway roof and proposed a cross-boundary anchor-grouting (CBAG) differential support technique. Calculations proved that the new support was particularly effective in restraining the expansion of tension cracks, thus preventing the slipping and dislocation deformations of rock masses on the curved roof side. The feedback of engineering applications showed that the maximum development depths of cracks in the arc roof and straight inclined roof of the roadway 150 m behind the working face are only 1.5 m and 1.10 m, decreasing by 61.3% and 47.6%, respectively, compared with the primary support. The proposed technology offers an overall thick-layer bearing structure for the surrounding rocks of roadways, effectively restraining the non-equilibrium large deformations of roadways in steeply inclined coal seams.

**Keywords:** steeply inclined coal seam; time-dependent failure; CBAG; differential support; equivalent anchorage

## 1. Introduction

China's proven reserves of steeply inclined coal seams account for about 20% of its total coal resources, but its annual mining volume only accounts for 4% of the total coal mining volume, showing huge mining potential of steeply inclined coal seams in the future [1]. Notably, in provinces such as Yunnan, Guizhou, and Sichuan in Southwest China, more than 50% of the coal seams are steeply inclined coal seams, and the mining of steeply inclined coal seams is crucial to the region's economy. Compared with flat coal seams or gently inclined coal seams, it is more difficult to control roadways in steeply inclined coal seams. Characterized by complex geological structures and unpredictable coal seam occurrences, such coal seams are prone to large deformations, destruction, and even roof collapses. A survey [2,3] shows that the repair rate of steeply inclined coal seam roadways exceeds 80%, and many roadways are scrapped due to improper maintenance. The main reason for this is that unreasonable roadway support parameters cause large deformations of surrounding rocks, severely hindering sustainable production. Therefore, the safety maintenance and control of roadways in steeply inclined coal seams have attracted more and more attention from experts and scholars.

The difficulties in controlling roadways in steeply inclined coal seams are that the self-stable performance of surrounding rocks is poor, and it is not easy to build a stable and reliable anchorage structure [4–6]. Three factors contribute to this situation: first, all of the occurrence angles of coal seams exceed 45°, and a coal–rock mass with a large dip angle will naturally be subject to slipping and dislocation deformations under the action of self-weight stress; second, most steeply inclined coal–rock masses have experienced multiple geological tectonic movements and have poor lithology as well as low strength; third, the thickness of a coal–rock seam varies greatly, the coal–rock interface is widely distributed, and the development of joints is obvious. However, the damage process and failure mechanisms of roadways in steeply inclined coal seams are still unclear, so the existing support system does not have a good control effect, and deformations as well as damage continue to occur.

Most roadways in underground coal mines are characterized by time-dependent deformations; that is, the deformations of surrounding rocks are due to the cumulative damage of coal–rock masses, which does not happen suddenly. Scholars study the time-dependent deformations of surrounding rocks mainly through numerical simulations, theoretical calculations, and field measurements [7,8]. The deformations and damage of roadway in steeply inclined coal seams are closely related to time; large deformations and damage of roadways often occur in later periods. Yang's analysis [9] shows that the gradual adjustment of surrounding rock stress is the main reason for large deformations of soft rock roadways with discrete elements. Sun [10] simulated the time-dependent deformation characteristics of roadways with FLAC 3D, and reduced the deformation speed of roadways by using jet grouting to impose constraints on surrounding rocks. Zhang [11] adopted concrete-filled steel tubular supports to passively control deep soft rock roadways and obtained deformation parameters of steel tubes via theoretical calculations to invert the deformation law of surrounding rocks. Tao [12] used a physical model to test the failure process of inclined layered soft rock roadways; through the real-time monitoring of the internal strain of the model during the failure process the failure mechanisms of inclined layered soft rock roadways are explored. On-site observation methods, such as borehole peeping, are the most intuitive means by which to study time-dependent failure of roadways, allowing clear observations of the number, angles, and openings of cracks in surrounding rocks for analyses of the evolution regularity of cracks in surrounding rocks of roadways over time [13–15].

In order to study the dynamic failure law and new control technologies of surrounding rocks of roadways in steeply inclined coal seams, this work studied the typical steeply inclined coal seam roadway of the Zhaojiaba Coal Mine in Sichuan Province, China. Methods such as numerical simulations, theoretical calculations, and field observations were employed to comprehensively analyze the time-dependent failure mechanisms of the roadway in the steeply inclined coal seam. A targeted CBAG differential support technology that has achieved a good control effect through field verification is proposed.

## 2. Asymmetric Deformation Characteristics of a Roadway in a Steeply Inclined Coal Seam

### 2.1. Basic Overview of the Roadway

Located in Guangyuan City, Sichuan Province, China, the Zhaojiaba Coal Mine, as shown in Figure 1a, is within the mining range of the Sichuan Coal Industry Group Limited Liability Company. The mine is a typical steeply inclined thin coal seam. At present, the main area undergoing mining is Mining Area 106, and the buried depth of the working face is 350–600 m. The roadway currently being excavated is the ventilation roadway of Working Face 1961, which is 15 m horizontally and 20 m vertically from the upper goaf. This roadway, a typical one in the steeply inclined coal seam, is excavated along two steeply inclined coal seams, i.e., 9# coal seam and 10# coal seam. The average dip angle of the two coal seams is 53°, the thickness of 9# coal seam is 1.1 m, and the thickness of 10# coal seam is 1.0 m, both of which are thin coal seams. The rock stratum between the two coal seams is carbonaceous mudstone with a thickness of 2.5 m. The ventilation roadway of

Working Face 1961 is 850 m long, and always crosses the two coal seams. The rock stratum histogram of the roadway is shown in Figure 1b.

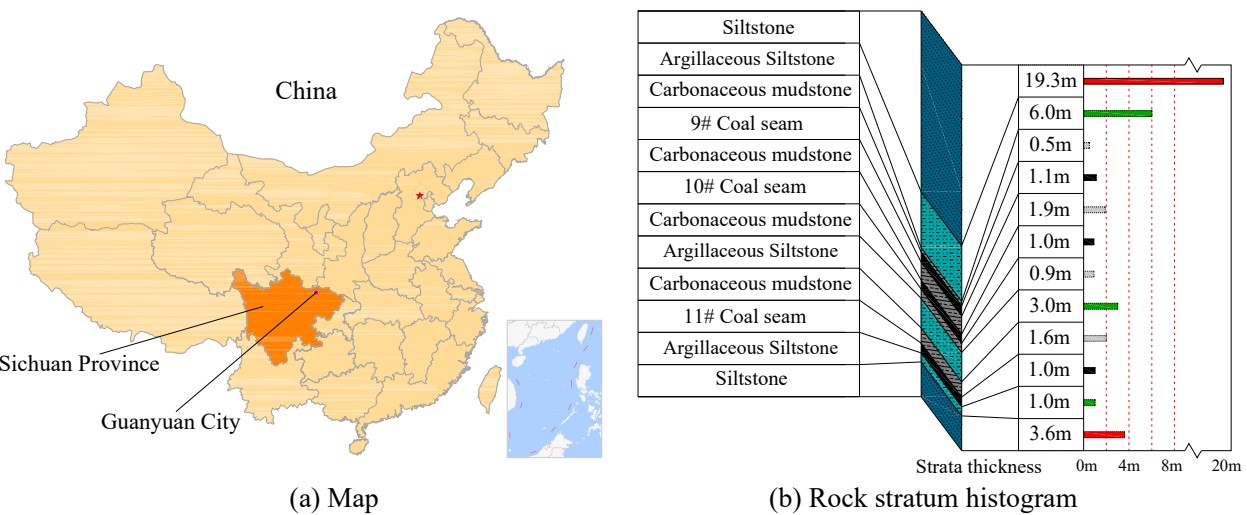

| (a) Map | (b) Rock stratum histogram |

**Figure 1.** Coal mine location and coal seam histogram.

### 2.2. Primary Support Scheme

According to the engineering experience, the ventilation roadway of Working Face 1961 adopts the vertical wall irregular top section, with the roadway width being 4.00 m and the roadway height being 3.55 m, as shown in Figure 2. The coal–rock mass is cut in parallel to the immediate roof of 9# coal seam without damaging the rock mass structure of the immediate roof, and the roof on this side is a straight inclined roof; the coal–rock mass is then cut perpendicularly or obliquely to 9# coal seam and the mudstone interlayer to form an arc roof, and the roof on this side is curved.

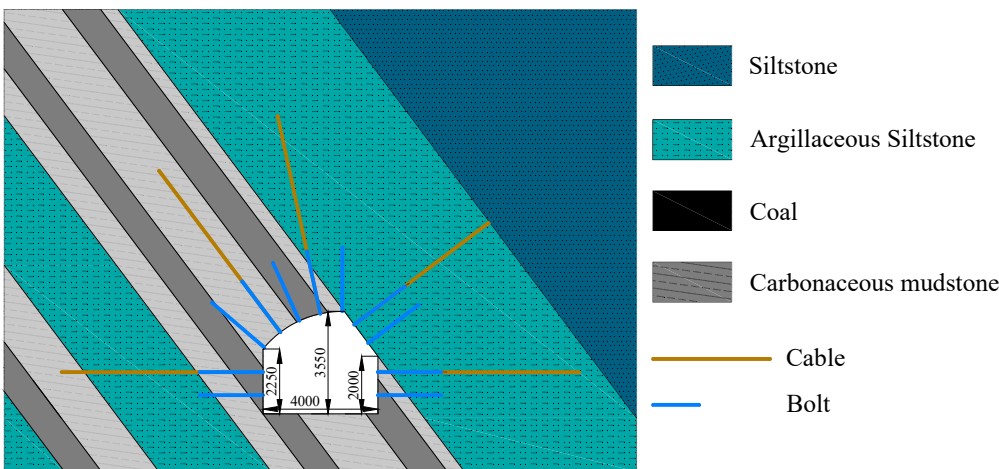

**Figure 2.** Irregular roadway section and primary support form.

As shown in Figure 2, the primary support of the ventilation roadway of Working Face 1961 is in the form of a bolt–anchor cable combined support, for which 11 bolts with a specification of Φ20 × 2200 mm are installed in each row, with the bolt spacing and row spacing being 800 mm × 800 mm and the bolts being connected by reinforced ladder beams. Five anchor cables with a specification of Φ17.8 × 7000 mm are installed in each row, with the anchor space and row spacing being 1600 mm × 1600 mm and the front as well as rear anchor cables of every two rows being supported by I-beams. The full section of the roadway is provided with a rhomboid metal mesh to protect its surface. Each bolt

is matched with two K2335 anchoring agents, and each anchor cable is matched with five K2335 resin cartridges. On-site measurement feedback shows that the pre-tightening force of the bolt in the primary support is only 30 kN and that the pre-tightening force of the anchor cable is only 80 kN.

### 2.3. Asymmetric Deformation Characteristics of the Roadway

Figure 3 shows the deformation of the ventilation roadway of Working Face 1961 under the primary support, which is 100 m away from the excavation face. It can be seen from the figure that different positions of the roadway show differential deformation characteristics: ① The rock mass between the two rows of bolts on the straight inclined roof experiences a "bulge" deformation, with the surface layer being relatively loose and layered shedding occurring locally; ② the curved roof experiences quite evident deformation, with the coal–rock seam being displaced and deformed, the surface arc shape becoming flat, and the reinforced ladder beams at this position being generally bent and locally broken; and ③ the corner roadway zone on the arc roof side is "extruded" as a whole, while the floor experiences a slope-type deformation (the closer it is to the left side, the greater the floor heave). To sum up, roadway deformations are characterized by asymmetric deformations and failure. The starting point, A, of the asymmetric line is located at the intersection of the arc roof and the straight inclined roof, while the ending point, B, is located at the intersection of 10# coal seam and the mudstone interlayer. The surrounding rock deformation in the zone to the left of the straight line, AB, is significantly greater than in the zone to the right of the straight line, AB. The continued deformation in the left side area caused a continuous reduction in the section of the roadway, with a maximum reduction proportion of 37%. Furthermore, the anchor cables tended to break in some areas, greatly undermining mining safety.

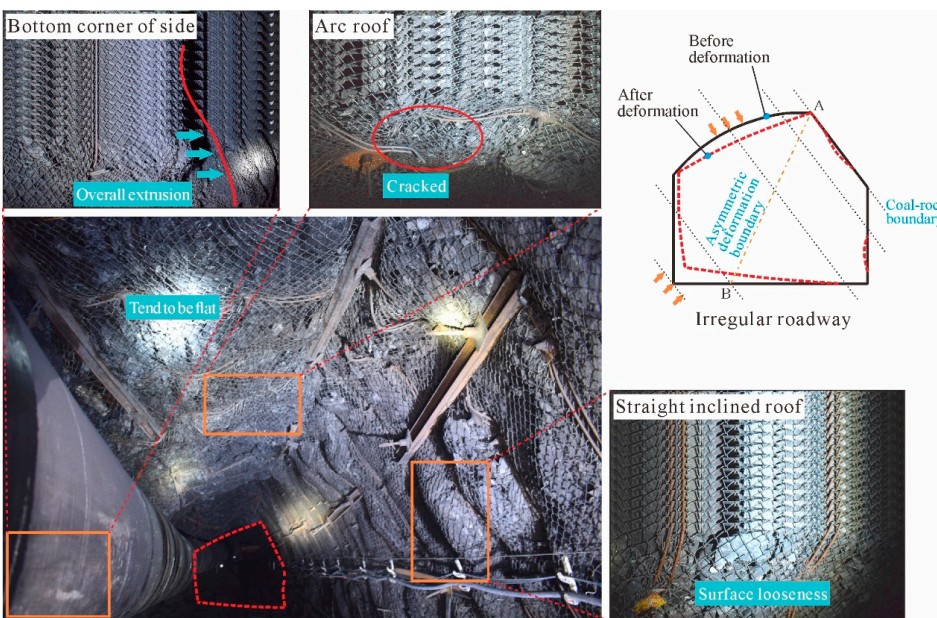

**Figure 3.** Failure status of the roadway under the primary support.

### 2.4. Evolution Characteristics of Cracks

In order to further evaluate the maintenance and control effects of the primary support, borehole peeping is carried out on the straight inclined roof and the arc roof at 4 m and 100 m away, respectively, from the excavation face to observe the development depths of initial cracks when the excavation begins, as well as to analyze the evolution of cracks in surrounding rocks after a period of roadway formation, as shown in Figure 4. The equipment used for the observation is a borehole peeping instrument, and the diameter and length of the borehole are 32 mm and 8 m, respectively.

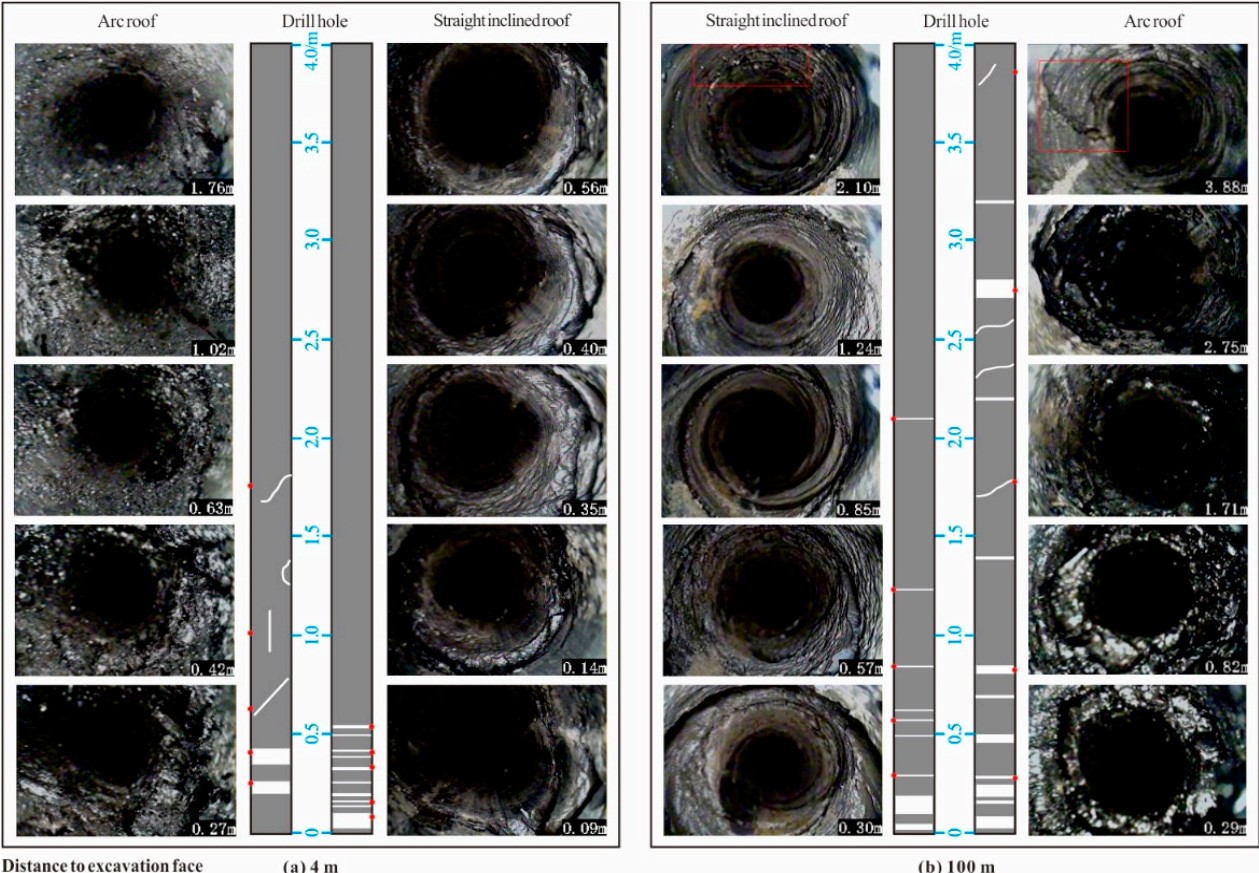

**Figure 4.** Distribution characteristics of cracks in the surrounding rocks of the roadway.

Figure 4a shows the distribution of cracks in the surrounding rocks when the roadway has just been excavated. At this time, the cracks' evolution depths at different roadway positions are obviously different. Cracks in the straight inclined roof are mainly within 0.5 m, and most of them are small annular cracks, with the maximum crack depth being only 0.56 m, while the coal–rock mass on the arc roof is of poor integrity, with the maximum crack depth being 1.76 m. In addition to annular cracks there are also inclined and longitudinal cracks in the hole. Compared with straight inclined roofs there are more cracks in the arc roof, and the cracks' openings are larger, undermining the arc roof's stability.

Figure 4b shows the distribution of cracks in the surrounding rocks 100 m away from the excavation face. Compared with when the excavation had just begun, the cracks on both sides of the surrounding rocks of the roadway expanded significantly at this time. The depths of the cracks in the straight inclined roof expand to 2.10 m, with a growth rate of 73.3%. It is additionally found from the figure that the cracks are mainly distributed within 1.5 m. The maximum development depths of the cracks in the arc roof reach 3.88 m, with an increase of 54.6%. The cracks and cracked zones are generally distributed within 3.0 m. When 100 m away from the excavation face the excavation stress tends to be stable, the cracks in the straight inclined roof expand to the bolt anchorage end area, and the cracks in the arc roof expand obviously beyond the bolt anchorage range, indicating that the shallow rock mass of the roadway as well as the bolt have experienced synchronous deformation and that the reinforcement effect of the bolt on the rock mass is very limited. Moreover, the expanding rate of the anchor cable is generally lower than 3.5%, and continuous rock mass deformation forces the anchor cable to break, such that the primary support can no longer meet the safety maintenance and control requirements of the roadway.

**3. Time-Dependent Failure Mechanisms of a Roadway in a Steeply Inclined Coal Seam**

*3.1. Model Establishment and Scheme Design*

3.1.1. Determination of Key Parameters

In order to analyze the differential damage and failure process of the roadway in the steeply inclined coal seam, the UDEC (universal distinct element code) numerical software is used to carry out a simulation study. The UDEC is a discontinuous mechanics method that uses discrete element method theory to provide accurate analyses for geotechnical engineering and mining engineering. It simulates the mechanical response of a jointed rock stratum under static or dynamic load, such as stress, displacement, and crack evolution [16,17]. The Trigon module in the software can realistically simulate the generation, expansion, and coalescence of cracks in brittle materials [18,19]. It was developed by Gao and Stead [20] on the basis of the Voronoi algorithm, which mainly divides a randomly generated polygonal mesh into triangular blocks. The resulting model has less dependence on a mesh, improves the problem of the post-peak strain hardening of rocks, and can realistically simulate fracturation as well as crack evolution in mining engineering.

In the discrete element model the mechanical parameters of the block and contact surface jointly determine the mechanical properties of a rock mass. The parameters of the block include density and the bulk as well as shear moduli, while those of the contact surface include normal stiffness, tangential stiffness, cohesion, and internal friction angle. The bulk modulus, $K$, and shear modulus, $G$, in the model are determined by the elastic modulus, $E$, and Poisson's ratio, $\nu$; the specific conversion relationship (see Equations (1) and (2)) is as follows [21]:

$$K = \frac{E}{3(1 - 2\nu)} \tag{1}$$

$$G = \frac{E}{2(1 + \nu)}, \tag{2}$$

In the Trigon module the elastic modulus of the triangular block depends on the normal stiffness and tangential stiffness of the contact surface of the block, and its calculation formula (see Equations (3) and (4)) is as follows [22]:

$$k_n = 10 \left[ \frac{K + \frac{4}{3}G}{\Delta Z_{\min}} \right], \tag{3}$$

$$k_s = 0.4 k_n, \tag{4}$$

where $k_n$ is the normal stiffness of the contact surface, $k_s$ is the tangential stiffness of the contact surface, and $\Delta Z_{\min}$ is the minimum side length of the block.

Therefore, in order to determine the mechanical parameters of the model block and the contact surface, it is necessary to obtain those of the real rock stratum in combination with laboratory experiments to verify the rationality of the model parameters.

1. Determination of Rock Mass Parameters

The parameters of coal–rock samples, i.e., rock parameters, are obtained through experiments, while the parameters applied in UDEC are rock mass parameters; therefore, the conversion of mechanical parameters is required.

Zhang and Einstein [23] proposed the conversion formula between the elastic modulus of rock mass and the elastic modulus of rock, as shown in Equation (5):

$$\frac{E_m}{E_r} = 10^{0.0186 RQD - 1.91} \tag{5}$$

where $E_r$ is the rock elastic modulus; $E_m$ is the rock mass elastic modulus; and $RQD$ is the rock quality index, which is obtained by peeping from a borehole in the ventilation roadway of Working Face 1961.

Singh and Seshagiri [24] found that there is a strong linear relationship between the $n$th power of the ratio of the rock mass elastic modulus to the rock elastic modulus and the ratio of the rock mass compressive strength to the rock compressive strength, as shown in Equation (6):

$$\frac{\sigma_{cm}}{\sigma_c} = \left(\frac{E_m}{E_r}\right)^j, \tag{6}$$

where $\sigma_c$ is the rock compressive strength; $\sigma_{cm}$ is the rock mass compressive strength; and the general value of $j$ is 0.56.

The rock mass tensile strength is shown in Equation (7):

$$\sigma_{tm} = k\sigma_{cm}, \tag{7}$$

Hoek and Brown [25] proved that the value of $k$ is generally 0.05–0.1, so its value is 0.1 this time.

The rock mechanical parameters are converted into rock mass mechanical parameters by calculation and filled into Table 1.

**Table 1.** Conversion of rock parameters and rock mass parameters.

| Rock Stratum Lithology | Rock Mechanics Parameters | | RQD | Rock Mass Mechanical Parameters | | |
|---|---|---|---|---|---|---|
| | $\sigma_c$/MPa | $E_r$/GPa | | $\sigma_{cm}$/MPa | $E_m$/GPa | $\sigma_{tm}$/MPa |
| Coal | 7.9 | 1.4 | 63 | 1.2 | 0.3 | 0.1 |
| Muddy siltstone | 25.2 | 6.4 | 82 | 15.3 | 2.6 | 1.5 |
| Carbonaceous mudstone | 12.3 | 3.1 | 76 | 6.5 | 1.0 | 0.8 |
| Siltstone | 37.8 | 14.5 | 90 | 27.9 | 8.4 | 2.8 |
| Fine-grained sandstone | 40.5 | 16.2 | 95 | 33.7 | 11.7 | 3.4 |

2. Verification of Model Parameters

The micro-parameters of the contact surface are calculated with rock mass parameters according to Equations (1) to (4). A $5 \times 10$ m plane model is established by the UDEC to carry out uniaxial compression simulation experiments and obtain simulated mechanical parameters, such as the compressive strength and elastic modulus of each rock stratum. The obtained simulated values are compared with the experimental values. If the error between the simulated and experimental values is large, it is necessary to adjust the rock stratum parameters again for another simulation, and so on.

After many simulation adjustments, the microscopic parameters with an error of less than 5% are set as the final mechanical parameters of the model, as shown in Table 2.

**Table 2.** Mechanical parameters of each rock stratum in the model.

| Rock Stratum Lithology | Block Parameters | | Contact Surface Parameters | | | | |
|---|---|---|---|---|---|---|---|
| | Density $\rho$/kg·m$^{-3}$ | Elastic Modulus $E_m$/GPa | Normal Stiffness $k_n$/GPa·m$^{-1}$ | Tangential Stiffness $k_s$/GPa·m$^{-1}$ | Internal Friction Angle $\theta$/° | Cohesion $C$/MPa | Tensile Strength/MPa |
| Coal | 1350 | 0.3 | 16 | 7 | 10 | 1.0 | 0.1 |
| Muddy siltstone | 2180 | 2.6 | 53 | 21 | 12 | 4.1 | 1.5 |
| Carbonaceous mudstone | 2060 | 1.0 | 47 | 19 | 12 | 2.3 | 0.8 |
| Siltstone | 2300 | 8.4 | 109 | 44 | 16 | 8.7 | 2.8 |
| Fine grained sandstone | 2550 | 11.7 | 79 | 32 | 19 | 11.9 | 3.4 |

3.1.2. Model Establishment

The UDEC model is designed according to the geological occurrence of the ventilation roadway of Working Face 1961, and the model size is 50 m × 50 m. It can be seen from Figure 5 that the model is divided into twenty layers, including five lithologies, namely coal, argillaceous siltstone, carbonaceous mudstone, siltstone, and fine-grained sandstone, and that the dip angle of each rock stratum is set to 53°. The model adopts the mechanical parameters calibrated in Section 3.1.1, as shown in Table 2. The origin of the model is set at the lower-left corner, and the surrounding zone of the roadway (the red part in Figure 5) is the key observation zone. The rock mass joints in this zone are densified in order to observe the distribution of cracks in the surrounding rocks.

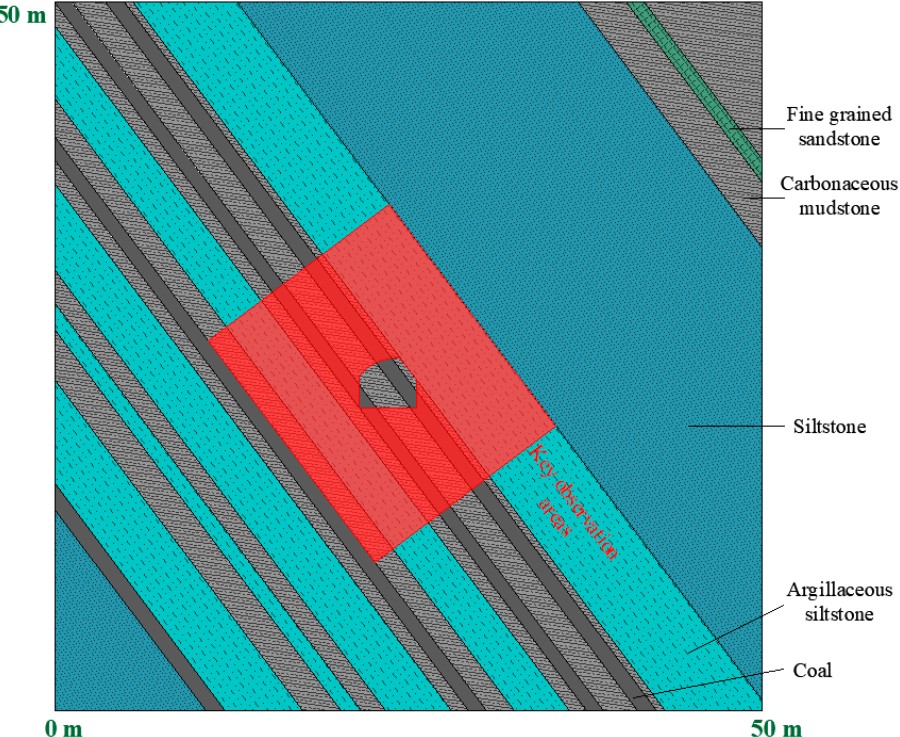

**Figure 5.** Model layout.

According to the geological data, the average burial depth of the ventilation roadway of Working Face 1961 is 450 m. If the lateral pressure coefficient is taken as 1, the vertical stress and horizontal stress applied by the model are both 11.76 MPa. The model is a plane strain model, the block is an elastic body, and the contact surface adopts the Mohr–Coulomb yield criterion. When the roadway module is deleted, this indicates that the excavation of the roadway has begun.

The cutting equipment for the ventilation roadway of Working Face 1961 is an EBZ260 TBM. Due to the limitation of the surrounding rocks around the roadway, the surrounding rock stress is reduced gradually rather than completely released at one time [26], so the Fish function "ZONK.FIS" (Itasca, 2012) in the UDEC is used to simulate the real process of stress release after excavation unloading of the roadway. The surface stress of the roadway is gradually released in ten stages, namely Stages ①, ②, ③, ④, ⑤, ⑥, ⑦, ⑧, ⑨, and ⑩. Of the surface stress, 10% is released in each stage until it drops to zero by Stage ⑩. The stress release coefficient is set as R, and the R of these ten stages is 0.1, 0.2, 0.3 . . . ~1.0, respectively, as shown in Figure 6. Four operation steps, namely four stress states, are selected to analyze the time-dependent failure law of the roadway. The operation steps are a (0.2 × 104), b (2.5 × 104), c (5.0 × 104), and d (10.0 × 104), and the stress release coefficients are 0.1, 0.7, 1.0, and 1.0, respectively.

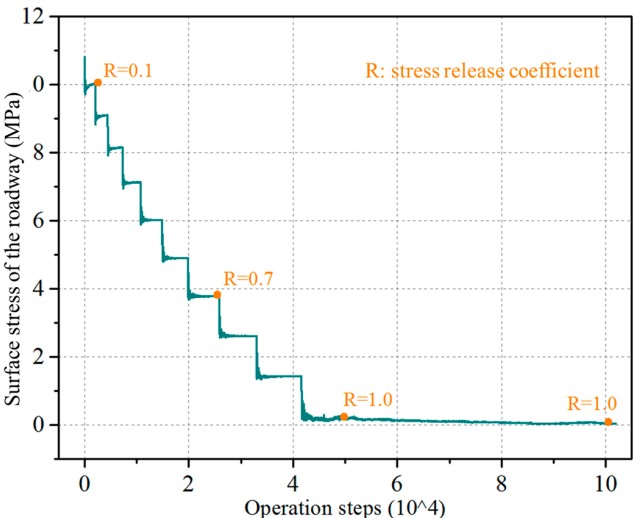

**Figure 6.** Surface stress evolution curve of the roadway.

### 3.2. Time-Dependent Failure Process of a Roadway in a Steeply Inclined Coal Seam

### 3.2.1. Stress Evolution Analysis

Figure 7 shows the nephogram of the maximum principal stress of the four stages in the stress release process of the roadway in the steeply inclined coal seam. It can be seen from the figure that after the excavation of the roadway is started the floor is damaged first, and its stress decreases accordingly. The nephogram of the maximum principal stress is in the shape of an "inverted triangle". The stress of the floor zone on the arc roof side decreases the most, with a relatively wide damage range, and the stress of the wall on this side also decreases in a small range. It is also found that stresses are concentrated in the argillaceous siltstone seam at the lower left of the roadway. With the gradual release of surrounding rock stress the damage degree of the surrounding rocks on the straight inclined roof side is small, and the degree of stress decreasing is not obvious; however, the damage to the surrounding rocks on the arc roof side becomes more and more serious, and the degree of stress decreases becomes increasingly obvious, which further causes the stresses to concentrate in the hard rock stratum at the lower left of the roadway. At this time, the roadway is at risk of being extruded from the lower left.

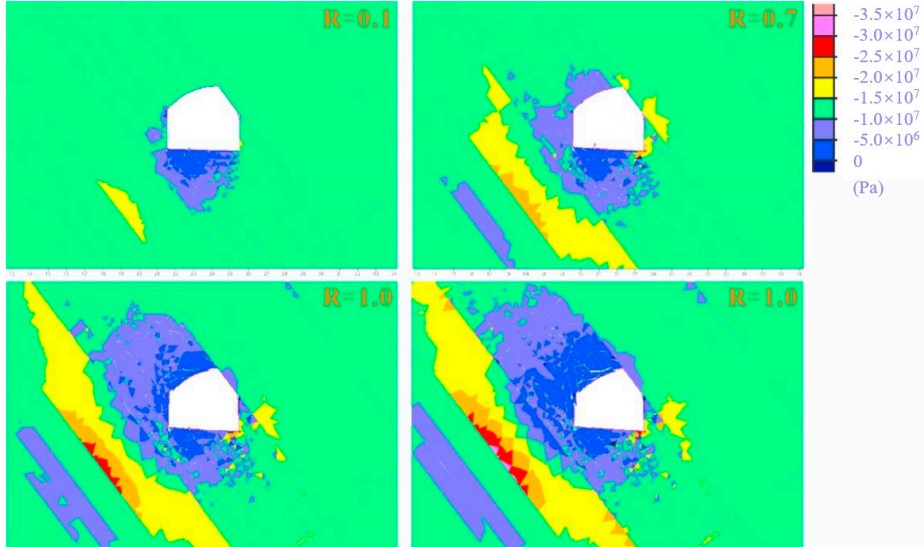

**Figure 7.** Nephogram of the maximum principal stress of the roadway during the stress release process.

### 3.2.2. Displacement Evolution Analysis

Figures 8 and 9 are the evolution diagrams of the surrounding rock displacement field of the roadway in the steeply inclined coal seam. It can be seen from Figure 8 that when the stress release coefficient, R, is 0.7, the coal–rock mass on the left side of the roadway floor is displaced upward, and the whole floor experiences a typical "slope-type" deformation with a large deformation on the left side and a small deformation on the right side. When the stress release coefficient, R, is 1.0, the coal–rock mass seam of the arc roof slips obviously and experiences a "falling" deformation, and the arc roof surface tends to be flat. It can be further concluded from Figure 9 that when the number of operation steps reaches $10 \times 104$ the maximum subsidence of the arc roof exceeds 300 mm, and the horizontal displacement at the left shoulder of the roadway is the largest, exceeding 200 mm. At the same time, the maximum floor heave of the roadway is located in the left corner of the roadway, and the deformation also exceeds 200 mm; however, the surrounding rock displacement on the straight inclined roof side is less than 50 mm. Therefore, the surrounding rocks of the roadway are steeply inclined coal seams that experience significant asymmetric differential deformations, the surrounding rocks on the arc roof side are a severely deteriorated zone, and the surrounding rocks on the straight inclined roof side is a slightly damaged zone.

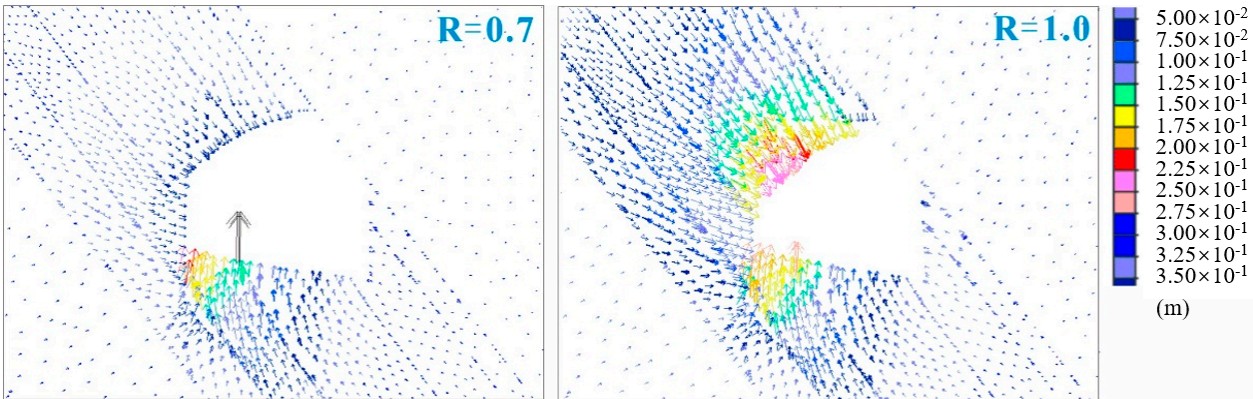

**Figure 8.** Vector diagram of the displacement field of the roadway.

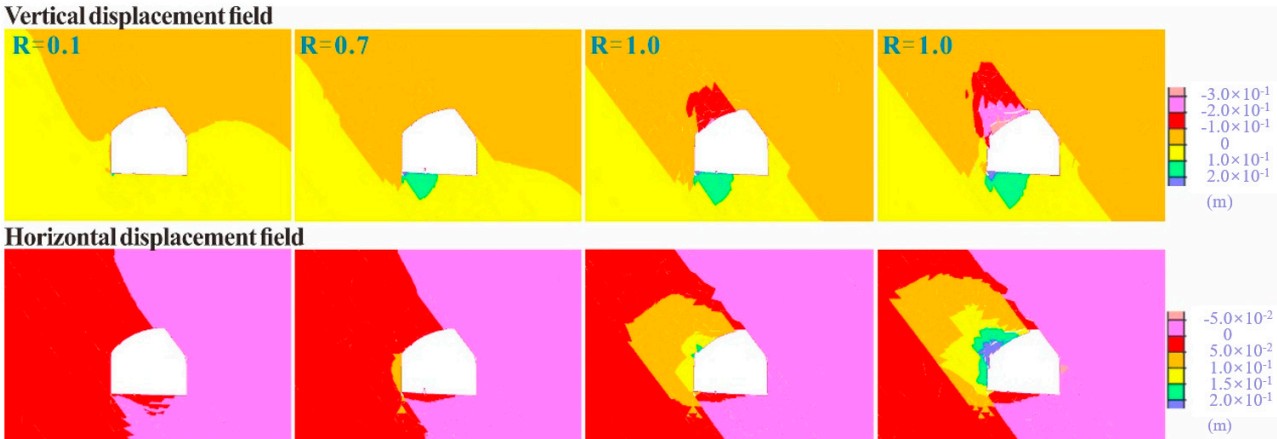

**Figure 9.** Nephogram of the displacement field of the surrounding rocks of the roadway.

### 3.2.3. Analysis of the Crack Evolution Law

Figure 10 shows the evolution characteristics of cracks in the surrounding rocks of the roadway in the steeply inclined coal seam. It can be seen from Figure 10a that the cracks in the surrounding rock are distributed in different zones, and that the number as well as evolution depths of the cracks in the surrounding rock zone on the arc roof side (Zone A) are significantly higher than those on the straight inclined roof side (Zone B).

Furthermore, Figure 10b shows that the cracks in the surrounding rocks are subject to the zonal evolution law. With the gradual release of surrounding rock stress, the cracks in Zone A increase significantly in stages. In particular, when R is 1 the number of cracks increases in a leaping manner; with the increase in calculation steps the number of cracks continues to increase, indicating that the deformation in Zone A has the property of long-term flowing deformation. However, the evolution trend of cracks in Zone B is significantly weaker than in Zone A, and the stability period is also significantly shortened. When the number of operation steps reaches $5 \times 10^4$, the number of cracks tends to stabilize. It can be found from the figure that the number of shear cracks in the two zones of the roadway is significantly larger than the number of tension cracks.

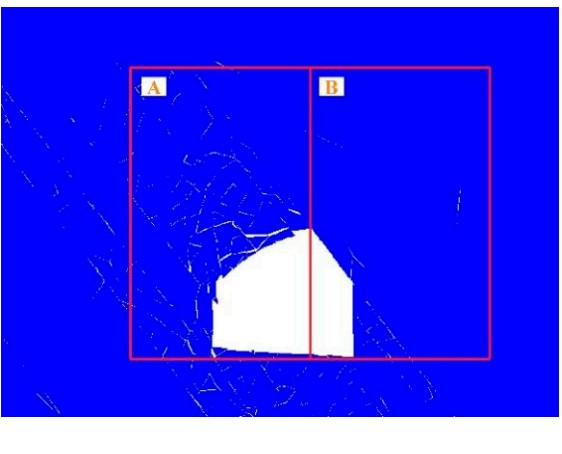

(**a**)

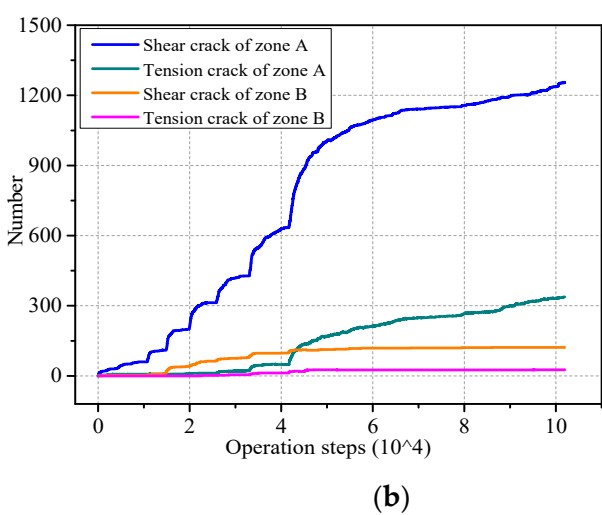

(**b**)

**Figure 10.** Differential evolution laws of cracks in surrounding rocks. (**a**) Zonal distribution characteristics of cracks; (**b**) time-dependent evolution curves of cracks.

3.2.4. Analysis of Time-Dependent Failure Mechanism

It can be concluded from the simulation results and field observations that the deformations of the surrounding rocks of the roadway in the steeply inclined coal seam are highly correlated with time and subject to the zonal differential evolution law, which is mainly reflected in the deterioration rate of the surrounding rocks on the arc roof side far exceeding that on the straight inclined roof side. The reasons for the time-dependent failure of the roadway are analyzed from the following two aspects:

(1) Fragmentation of rock mass in the floor corner on the arc roof side of the roadway in the steeply inclined coal seam is the precondition for the time-dependent failure of the surrounding rocks. It can be seen from Figure 8 that, after the excavation of the roadway is begun, the coal–rock mass in the floor corner zone first experiences an upward seam displacement deformation and that the damage degree at this position is the most serious, which provides deformation space for the later slipping failure of the surrounding rock on the arc roof side. The roadway floor also experiences a "slope-type" deformation with a large deformation on the left side and a small deformation on the right side. In order to realize the long-term maintenance and control of the roadway in the steeply inclined coal seam, it is necessary to control key parts of the floor on the arc roof side.

(2) Non-equilibrium deformations of the roadway in the steeply inclined coal seam are caused by surrounding rock stress. The occurrence of the steeply inclined coal–rock seam determines the deflection transfer of surrounding rock stress during the excavation process. Such asymmetric adjustment causes slipping and dislocation deformations of the coal–rock mass on the arc roof side along the seam, and the large-scale linkage of surrounding rocks tends to form a partial stress-bearing arch on the roadway, which intensifies the damage of the surrounding rocks on the peak stress side and structural instability damage. It can be seen from Figure 7 that the maximum principal stress is transferred to the hard rock stratum

deeper in the surrounding rocks, and the damaged as well as deteriorated parts further expand to the depth. The surrounding rock stress and the rock mass deformations have a bidirectional deterioration relationship, which leads to the failure of the bolt anchorage structure as well as the instability and collapse of the severely deteriorated zone; however, there is no large-scale stress concentration in the surrounding rock on the straight inclined roof side, and only a certain degree of damage occurs in the shallow part. Therefore, the floor angle of the roadway in the steeply inclined coal seam shall be timely controlled, and differential control measures shall be taken for surrounding rocks on both sides.

## 4. CBAG Differential Support Technology

### 4.1. Distribution Characteristics of Cracks in the Surrounding Rocks under the Combined Support of Short Bolts and Long Anchor Cables

The simulated roadway is controlled according to the original combined support of the mine in Figure 2 with the combination of 14 short bolts, with a length of 2.2 m, and 5 long anchor cables, with a length of 7.0 m. With the gradual release of the surface stress of the roadway in stages, the evolution characteristics of cracks in the surrounding rocks with the number of operation steps are shown in Figure 11. When the number of operation steps reaches 2.5 × 104, that is, R = 0.7, the number of cracks in the surrounding rocks in Zone A of the roadway is 257, while the number of cracks in the surrounding rocks in Zone B is only 43, indicating that the surface stress has not been completely released, such that the damage degree of the rock mass shows a great difference. It can be further seen from the crack distribution diagram that the maximum depth of crack evolution in Zone A is still in the bolt anchorage area, while the cracks in Zone B are only distributed on the shallow surface of the roadway. When the number of operation steps reaches 10 × 104, that is, R = 1.0, the number of cracks in the surrounding rocks in Zone A and Zone B is 794 and 117, respectively, and the cracks in different positions of the roadway increase to different extents. As shown in the lower-right figure, the cracks in Zone A expand significantly, and the evolution depth exceeds the bolt anchorage area, with the maximum crack depth reaching 3.5 m. At this time, the rock mass in the whole bolt anchorage area is seriously cracked, with the risk of overall collapse and instability.

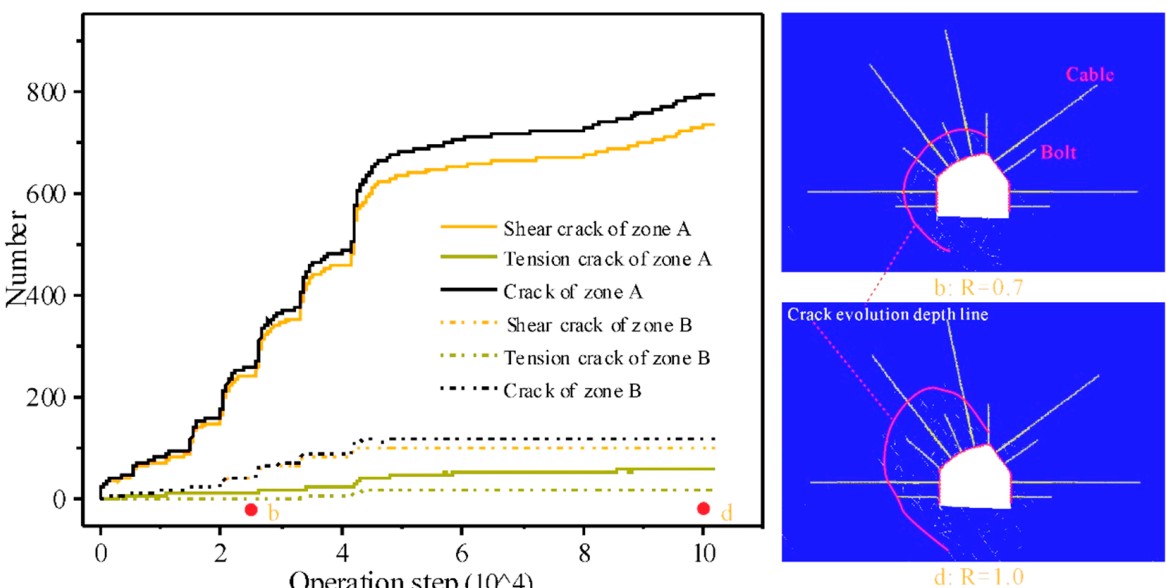

**Figure 11.** Evolution characteristics of cracks in the surrounding rocks under the primary support.

### 4.2. Thick-Layer Anchorage Principle and Differential Equivalent Reinforcement Technology

The number of cracks in the surrounding rocks in Zone A and Zone B of the roadway under the primary support of the mine is greatly reduced by 775 and 31, respectively,

compared with that when there is no support, with a reduction rate of 49.4% and 20.9%, respectively. Although the primary support has a certain effect on restraining the development of cracks, it does not change the damage status of the seriously deteriorated zone of the roadway (Zone A). The anchor point in the bolt on this side is still located in the crack circle, and the bolt anchorage zone shows an overall movement trend; however, since the anchor cable is too long it has a limited reinforcement effect on the rock mass in this range, and the anchor cable is easy to break. Therefore, the bolt and anchor cable combined technology needs to be changed urgently.

It is assumed that the anchored rock beam of the foundation of the roadway roof is simplified as a simply supported beam, as shown in Figure 12. When the rock beam experiences pure bending deformation, according to the maximum tensile stress theory (the first strength theory), the maximum dangerous point of the roof is at the center of the anchored rock beam surface (i.e., Point C). At this time, the maximum tensile stress of the rock beam (see Equation (8)) is as follows:

$$\sigma_{max} = \frac{M_{max}y_{max}}{I_Z}, \tag{8}$$

where $M_{max} = \frac{qL^2}{8}$, $y_{max} = \frac{h}{2}$, and $I_Z = \frac{bh^3}{12}$. The relationship (see Equation (9); $k$ stands for constant) between the maximum tensile stress and the thickness of the rock beam is further obtained as follows:

$$\sigma_{max} = \frac{3qL^2}{4bh^2} \Rightarrow k\frac{1}{h^2} \propto \frac{1}{h^2}, \tag{9}$$

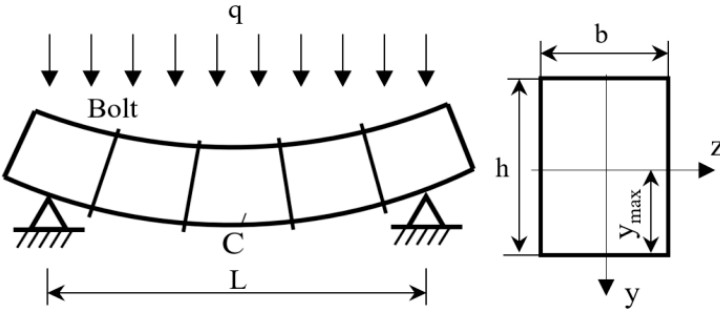

**Figure 12.** Mechanical model of a simply supported beam for an anchored rock beam.

It can be concluded from Equation (9) that the maximum tensile stress of an anchored rock beam is inversely proportional to the square of its thickness, indicating that the greater the thickness of an anchored rock beam the smaller the tensile stress value and the more stable the roof rock stratum. Therefore, in order to realize the long-term stable control of the roadway, the foundation anchorage thickness in the support system shall meet the cross-boundary requirements of the thick layer [27,28]; that is, the foundation bolt support shall span the surrounding rock crack circle as well as plastic circle, and the inner anchorage point shall be anchored into the elastic circle to form a thick reinforcement circle that is closely connected with the stable rock mass in the deep part. In order to improve the strength and stiffness of the anchorage bearing layer and resist the disturbance of mining stress as well as the long-term creep effect of soft rock, the bidirectional linkage control of the deep and shallow displacement is realized by increasing the thickness of the foundation anchorage layer and restricting the large displacement in the shallow part with the small displacement in the deep part, which is suitable for a roadway under any surrounding rock inclination angle.

According to the peep images and numerical simulation results of the surrounding rocks of the roadway, the initial cracks in the surrounding rocks of the roadway in the steeply inclined coal seam after excavation are characterized by an obvious asymmetric distribution. The crack depth in Zone A is generally greater than 1.5 m, while the crack

depth in Zone B is generally less than 0.5 m, as shown in Figure 13a; however, the equal-length and equal-strength combined support form of the primary support is no longer suitable, and the key positions are not reinforced by deep anchorage, so the maintenance and control effects of the roadway are extremely poor. According to the cross-boundary anchorage principle, the bolt–anchorage thickness of the severely deteriorated Zone A shall be significantly greater than that of the slightly damaged Zone B. At the same time, due to the repeated staggered distribution of the steeply inclined coal–rock seams in Zone A, in order to ensure the long-term bearing of the anchored rock mass, the grouting method shall be adopted to modify the lithology of the surrounding rocks, especially the grouting reconstruction of the rock mass in the floor corner. Unequal-length differential anchor-grouting support technology is used to construct the overall thick-layer bearing structure of the full-section surrounding rocks of the roadway, that is, the equivalent reinforcement layer, to adapt to the long-term creep effect of the weak surrounding rocks, and then realize the long-term stable bearing of the roadway in the steeply inclined coal seam, as shown in Figure 13b.

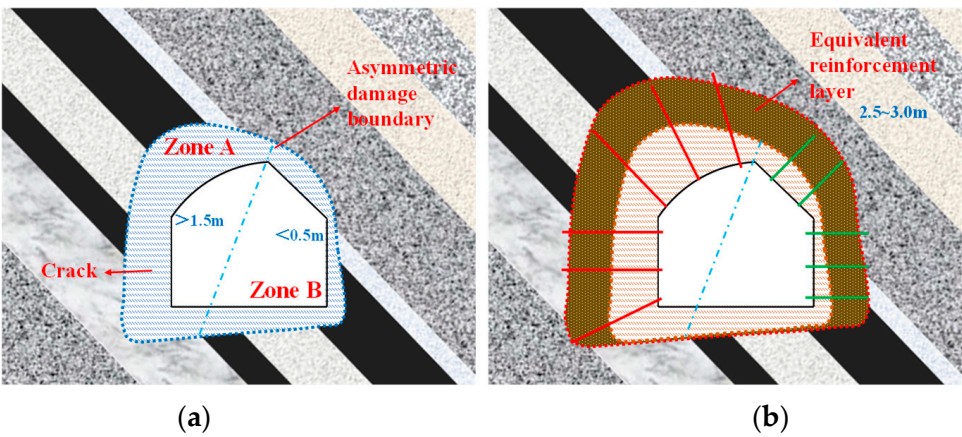

**Figure 13.** Asymmetric deterioration characteristics and equivalent reinforcement principle of the roadway in the steeply inclined coal seam. (**a**) Sketch of the asymmetric distribution of cracks; (**b**) equivalent reinforcement effect.

In view of different coal–rock properties, crack characteristics, and crack depths in each part of the roadway in the steeply sloping coal seam, two support forms, namely hollow grouting cables and flexible bolts, are used in Zone A and Zone B, respectively, to construct an equivalent reinforcement layer with a thickness of 2.5–3.0 m for the roadway in the steeply inclined coal seam. For the surrounding rocks in Zone A the anchor-grouting combined reinforcement method shall be adopted, and the foundation anchorage thickness shall exceed 4.0 m. For Zone B only the end anchorage is required, and the foundation anchorage thickness can be less than 3.5 m.

## 5. Engineering Case Analysis

### 5.1. Differential Support Scheme

Taking the intersection of the arc roof and the straight inclined roof of the irregular roadway as the dividing point, as shown in Figure 14, seven anchor cables with a length of 4300 mm and a diameter of 22 mm are installed on the left side of the roadway, with the red ones representing hollow grouting cables and the blue ones representing solid cables; four flexible bolts with a length of 3500 mm and a diameter of 22 mm are installed on the right side of the roadway. Both the anchor cables and flexible rods are matched with the 300 mm × 300 mm × 14 mm arched steel tray, and a reinforced ladder beam as well as a woven mesh are used to strengthen and protect the surface. The initial pre-tightening force of the anchor cable shall not be less than 120 kN, and the initial pre-tightening force of the flexible bolt shall not be less than 80 kN. The grouting material selection

of Huainan Donghuaouke company KWJG-1 type mineral reinforced composite mortar, a product without shrinkage, micro-expansion, high liquidity, and pumping, does not contain chloride. A ZBYSB40/22-7.5 mining hydraulic grouting pump was selected for grouting pumping.

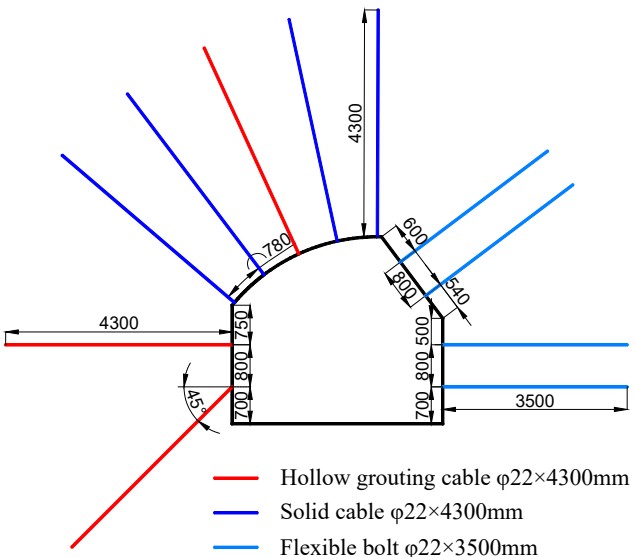

**Figure 14.** New support scheme.

It is required to note the following:① a long anchor cable with a size of Φ22 mm × 6500 mm shall also be installed for an arc roof area that is seriously cracked after the initial excavation; ② for serious floor heave, hollow grouting cables shall be installed at an inclination of 45° downward, and the deformations of the floor heave shall be suppressed by grouting coupling as well as the reinforcement of key parts of the floor corner.

### 5.2. Comparative Analysis of Simulation Effects of New Support

Figure 15 shows the characteristics and deformation of cracks in the surrounding rocks of the roadway under the new support when the number of operation steps reaches 10.0 × 104. Figure 15a shows that after the differential support is applied the number of cracks in the surrounding rocks of the roadway is significantly reduced, and that the distribution range is significantly narrowed. The above is specifically reflected in the following three aspects: ① the cracks on the arc roof side are all within the anchorage range, with the maximum evolution depth being reduced to 1.8 m, the cross-boundary support limiting the slipping as well as dislocation of the steeply inclined coal–rock mass, and the integrity of the surrounding rocks being significantly improved; ② the cracks on the straight inclined roof side are only distributed in the shallow part of the roadway, and the deformation is small; ③ and the number of floor cracks is also appropriately reduced, which is mainly because the 45° grouting anchor cable improves the shear capacity of the coal–rock mass in the floor corner, and the surrounding rock on the arc roof side is quite complete, which limits the deformations of the floor coal–rock mass as well as the development of cracks.

Figure 15b shows the evolution law of the displacement of the roadway's surrounding rocks with depths under different supports. The vertical line, EF, in the figure is the center line of the roadway. The distance between the horizontal line, GH, and the horizontal line, PQ, of the roadway floor is 2 m. A total of 40 measuring points are arranged vertically inward 5 m from points E, F, G, and H on the roadway's surface in order to obtain the deformation status of the roadway at different positions. The vertical displacement is monitored at each measuring point on the roof and floor, and the horizontal displacement is monitored at each measuring point on both sides. In addition, 40 measuring points are also arranged on the horizontal line, PQ, of the floor to observe the floor heave status.

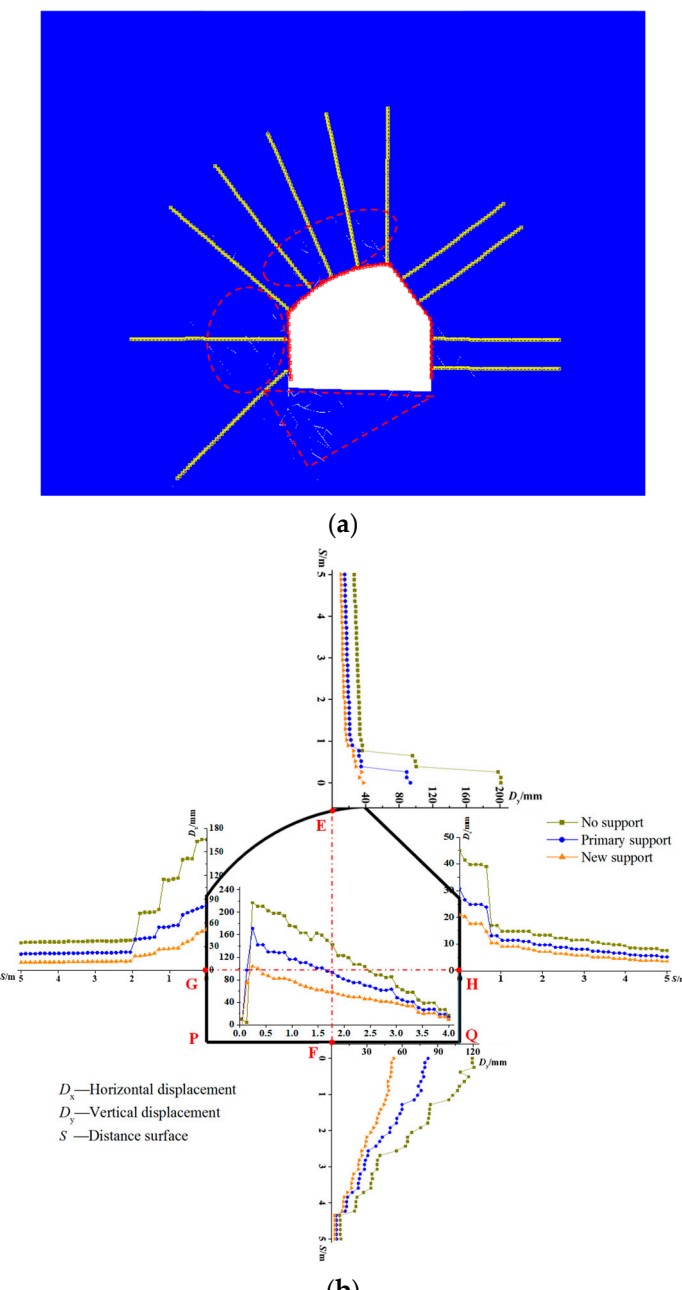

**Figure 15.** Maintenance and control effects of the roadway under the new support. (**a**) Distribution characteristics of cracks; (**b**) evolution law of the displacement of the roadway's surrounding rocks with depths under different supports.

It can be seen from the deformation status of the roadway at different positions that the control effects of different support forms are significantly different. It can be concluded from the subsidence data of the roof that deformation mainly occurs in shallow parts less than 1 m above the roof. The displacement of point E under no support, the primary support, and the new support is 205 mm, 98 mm, and 40 mm, respectively. The displacement under the new and primary supports decreases by 80.5% and 52.2%, respectively, compared with that under no support. It can be concluded from the displacement data of the wall on the arc roof side that deformations are mainly concentrated within 2 m of the wall. The displacement of point G under no support, the primary support, and the new support is 165 mm, 85 mm, and 50 mm, respectively. The displacement under the new and primary supports decreases by 69.7% and 48.5%, respectively, compared with that under no support.

It can be concluded from the displacement data of the wall on the straight inclined roof side that deformations are mainly concentrated within 0.8 m of the wall. The displacement of point H under no support, the primary support, and the new support are 45 mm, 31 mm, and 20 mm, respectively. The displacements under the new and primary supports decrease by 55.6% and 31.1%, respectively, compared with that under no support. It can be concluded from the displacement data of the floor that deformations are mainly concentrated within 4.2 m. The displacement of point F under no support, the primary support, and the new support is 120 mm, 81 mm, and 52 mm, respectively. The displacement under the new support and primary support decreases by 56.7% and 32.5%, respectively, compared with that under no support. It can be concluded from the displacement data of the horizontal line, PQ, of the floor that the maximum displacement point of the floor is 0.3 m away from point P as the origin, and it shows a single decreasing trend from 0.3 m to 4.0 m. The maximum floor heave under no support, the primary support, and the new support is 220 mm, 178 mm, and 102 mm, respectively. The maximum floor heave under the new support decreases by 53.6% and 19.1%, respectively, compared with that under the primary support and no support. It can be seen that the control effects of the new support at different positions are significantly better than those of the original support. The deformations under the new support decrease by 53.6–80.5% compared with those under no support, while the deformations under the primary support decrease by only 19.1–52.2% compared with those under no support.

*5.3. Mine Pressure Monitoring and Analysis*

The working load on anchor cables is monitored for a long time by the dynamometer clam between the tray and the nut, which has the functions of a digital display, historical data memory, and transmission. Figure 16 shows the evolution curve of the working load on anchor cables of the arc roof and the straight inclined roof under the new support. It can be seen from the figure that the evolution characteristics of the working load on anchor cables at different installation positions over time are quite different. It can be seen from Figure 16a that the stability period of the anchor cables is about 40 days, during which the load on the anchor cables is characterized by repeated oscillation up and down, with the maximum load being 168 kN and the minimum load being the pre-tightening force (123 kN), indicating that the coal–rock mass in the shallow part on the arc roof side is prone to damage and failure under compression, which will cause the tray to move and lead to a readjustment of the load on the anchor cables. After repeated mechanical action, balance is finally achieved and the load is stabilized, and the final load is about 161 kN. It can be seen from Figure 16b that the stability period of the flexible bolt is about 13 days, and its working load shows an evolutionary trend of a sharp increase at first, then a slow increase, and finally a stable bearing. The rock mass on this side has good integrity, and the lithology is relatively hard and flexible. There is no repeated oscillation or adjustment for the load on the flexible bolt. The initial pre-tightening force is 83 kN, and is finally stabilized at 165 kN. The characteristics of the load on the bolt–cable at different positions of the roadway also reflect differences in the properties of the rock mass, which confirms that the control methods also need to be differential.

Figure 17 shows the peeping images of the arc roof and straight inclined roof of the roadway. The cracks in the arc roof and the straight inclined roof are distributed within 1.50 m and 1.10 m, respectively. There are seven cracks and two cracked zones on the arc roof side, but only six small cracks on the straight inclined roof side. It can be concluded from the distribution of cracks in the surrounding rocks that the integrity of the rock mass is significantly improved, which is beneficial to the long-term stability of the roadway. Figure 18 is the site photo of the roadway after 150 m of excavation. The surfaces of the arc roof and the straight inclined roof of the roadway are very flat, and there are no deformations and damage similar to those that occur under the primary support.

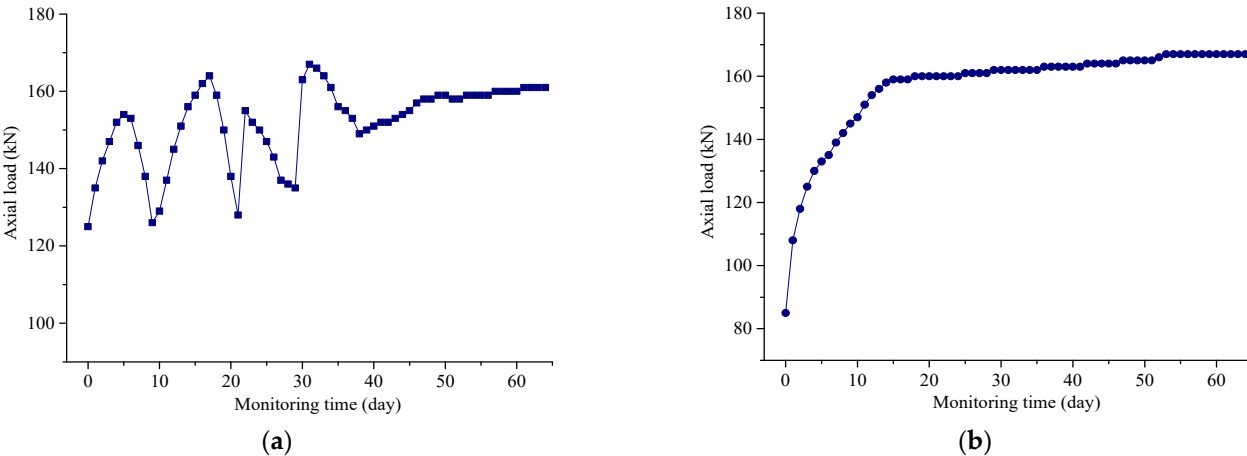

**Figure 16.** Curve of the working load on bolt/anchor cable under new support. (**a**) Arc roof; (**b**) Straight inclined roof.

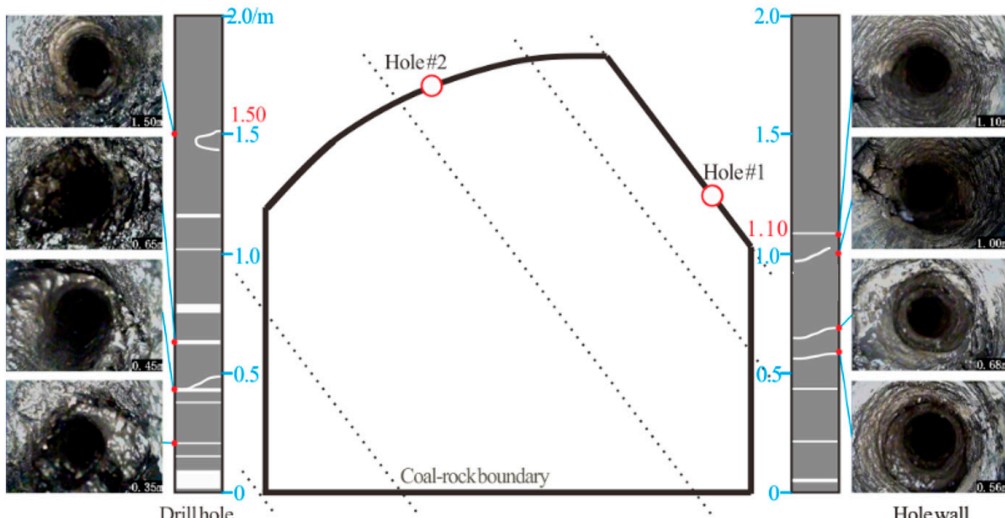

**Figure 17.** Peeping results of the surrounding rocks under the new support.

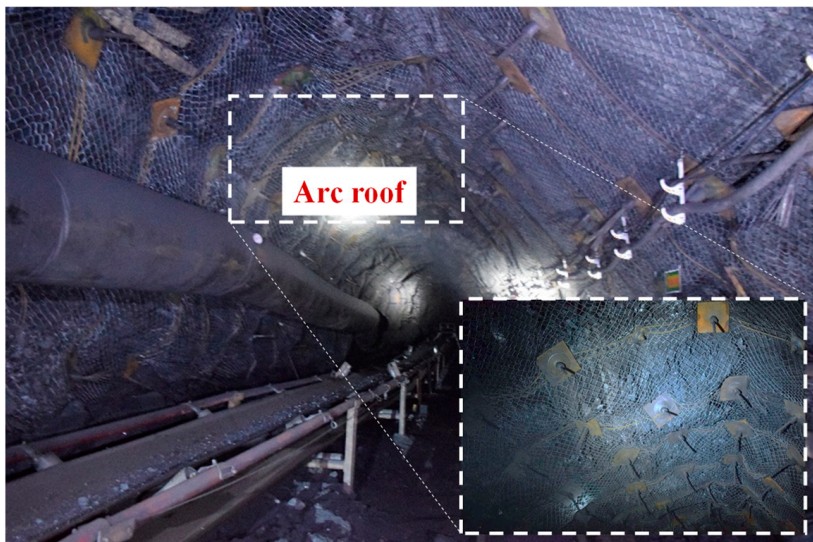

**Figure 18.** Maintenance and control effects of the roadway under the new support.

### 5.4. Discussion

5.4.1. Restraining Mechanism of Cross-Boundary Support on Cracks in the Surrounding Rocks

The selection of the roadway support in conditions of large deformations should consist in the common fulfillment of geometric, ventilation, and geomechanical criteria [29]. The new CBAG differential support technology improves the control effect of the irregular roadway. In order to further understand the difference in the control effect of the surrounding rocks under different support forms, Zone A, with the most obvious control effect, is selected for illustration. Figure 19 shows the evolution curves of cracks in Zone A of the irregular roadway under different support forms. It can be seen from Figure 19a that when the operation coefficient reaches $5 \times 104$, that is, the surface stress of the roadway is just released to zero, the number of shear cracks in Zone A under no support, the primary support, and the new support is 1005, 636, and 337, respectively. The number of shear cracks under the primary support and the new support decreases by 36.7% and 66.5%, respectively, compared to that under no support. At the same time, as the number of operation steps continues to increase, when the operation coefficient reaches $10 \times 104$ the number of shear cracks under no support, the primary support, and the new support increases by 23.1%, 15.1%, and 0.3%, respectively. It can be seen from Figure 19b that when the operation coefficient reaches $5 \times 104$ the number of tension cracks in the surrounding rocks of Zone A under no support, the primary support, and the new support is 171, 46, and 16, respectively. The number of tension cracks under the primary support and the new support decreases by 73.1% and 90.6%, respectively, compared with that under no support. When the operation coefficient reaches $10 \times 104$ the number of shear cracks under no support, the primary support, and the new support increases by 94.2%, 34.8%, and 0, respectively. It can be concluded from the data of shear cracks and tension cracks that as the number of operation steps continues to increase the evolution rate of tension cracks in the surrounding rocks under no support is significantly faster than that of shear cracks. At the same time, the CBAG new technology is used to reconstruct shallow surrounding rocks and bidirectional linkage control of deep and shallow rock masses, thus restraining the evolution of tension cracks and realizing the fundamental control of cracked surrounding rocks of irregular roadways. Therefore, restraining the expansion of tension cracks is the key to controlling the surrounding rocks of irregular roadways.

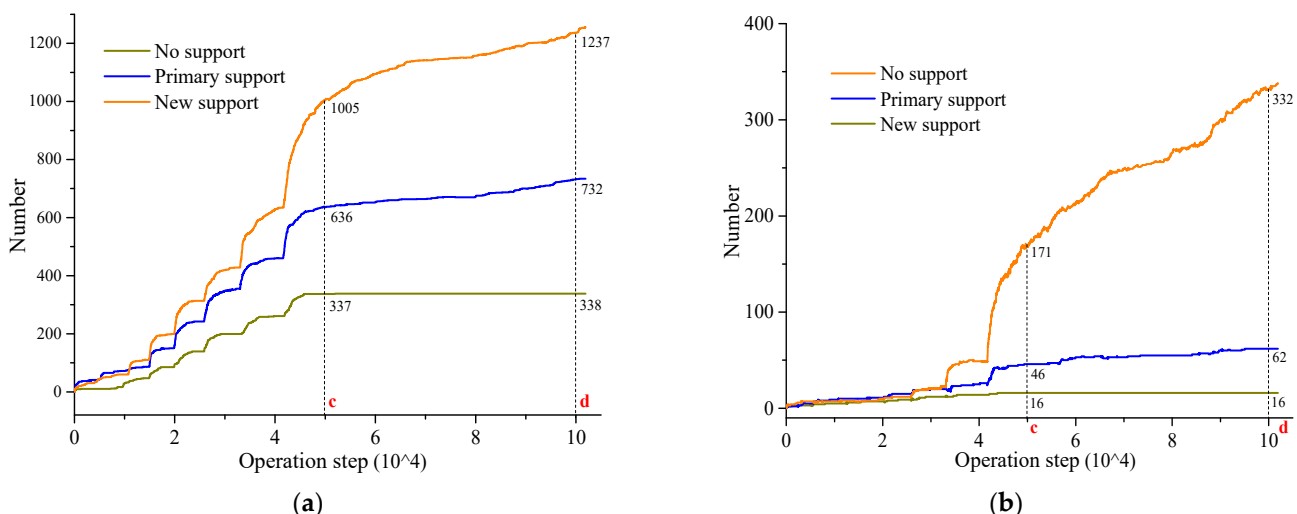

**Figure 19.** Evolution curves of cracks in Zone A of the roadway under different support forms. (**a**) Shear crack; (**b**) tension crack.

5.4.2. Discussion on CBAG New Support Structures and Grouting Sealing Method

Two new types of support products are adopted for the CBAG differential support, namely hollow grouting cables and flexible bolts, as shown in Figure 20. The hollow

grouting cables, with a specification of $1 \times 8$–22–2000 MPa, are produced by the fully automatic production line of Shandong Yanxin Company. They are mainly composed of eight high-strength steel wires and stainless steel pipes, which overcome the disadvantages of the traditional welded grouting anchor cable structure, realize the high strength of the anchor cable bolt body, and also change the traditional method of plugging the hole at the tail with a rubber plug. They use a hard pipe to push and squeeze the hose to make the hose expand, and then move the plugging position from shallow inside to the hole of the complete surrounding rocks, that is, from State ① to State ②, which overcomes the disadvantage of poor plugging caused by the cracked shallow part. Another product is flexible bolts with a specification of $1 \times 7$–18–1860 MPa. They are composed of a steel strand bolt body and a tail threaded sleeve. They overcome the disadvantage that the length of the threaded steel bolt is limited by the roadway space and cannot meet requirements of cross-boundary support, and use the nut rotation installation method to realize the rapid installation of the steel strand bolt body.

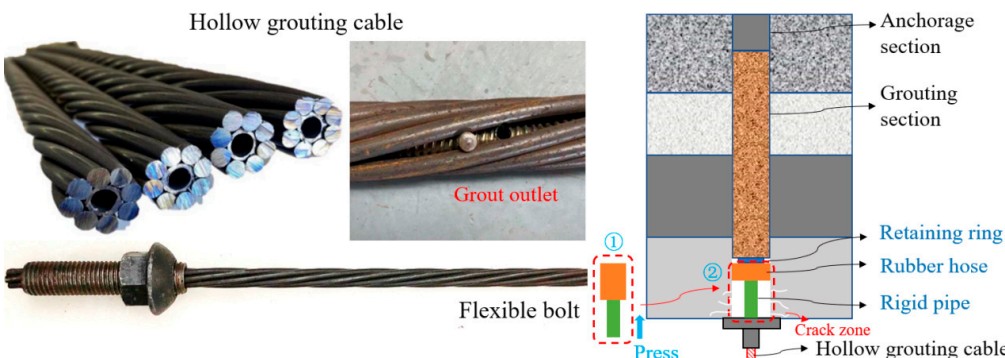

**Figure 20.** New support products and grouting sealing method.

## 6. Conclusions

(1) This work analyzes the asymmetric failure characteristics of the sliding and dislocation deformations of the coal–rock mass on the arc roof side and the loose deformations of the shallow rock mass on the straight inclined roof side of the roadway in a steeply inclined coal seam in Sichuan Province. The differential evolution regularities of cracks in the arc roof and the straight inclined roof during the initial and long-term excavation of the roadway are analyzed. The main reasons for the large deformations of the surrounding rocks are the coordinated deformations of the bolts, the shallow rock mass under the primary support, and the poor reinforcement of long anchor cables on the rock mass within this range.

(2) UDEC discrete element simulation was utilized to analyze the process stress evolution pattern. The result shows that stress release occurs first in the floor strata and then causes the gradual stress release of the surrounding rock on the entire arc roof side. Finally, the roadway shows a typical zonal failure characteristic of "slope-type" deformations for the floor, "dislocation falling" deformations for the arc roof, and micro-deformations for the straight inclined roof.

(3) Based on the non-equilibrium deformation characteristics of the roadway in the steeply inclined coal seam, this work proposes CBAG differential support technology for reinforcing key parts in the floor corner, thick-layer anchorage of the straight inclined roof and the arc roof, and grouting reconstruction. An equivalent thick-layer bearing structure was constructed for the full section of the surrounding rocks of the roadway. Using new hollow grouting cables and flexible bolts, this work anchored shallow grouting reconstructed rock mass into a deep rock stratum, such that a small displacement of deep rock mass can restrain a large displacement of shallow rock mass. In this way, it can adapt to the long-term creep effect of soft surrounding rocks of the roadway in the steeply inclined coal seam. Recommendation: The new grouted cable support should be adopted in all

roadways with similar conditions to the ventilation roadway in Working Face 1961 of the Zhaojiaba Mine.

(4) The analogue calculation shows that the new support greatly narrows the tensile stress zone of the surrounding rock of the roadway, effectively preventing slipping and dislocation deformation of the surrounding rock mass of the arc roof. On-site industrial testing shows that the depth of the maximum crack in the arc roof generally decreases by 61.3%, from 3.88 m to less than 1.50 m, and the development depth of cracks in the straight inclined roof decreases by 47.6%, showing a significant improvement in the surrounding rock control effect.

**Author Contributions:** Funding acquisition, Z.X. (Zhengzheng Xie) and N.Z.; Investigation, Z.X. (Zhengzheng Xie), J.W., N.Z., F.G., Z.H., Z.X. (Zhe Xiang) and C.Z.; Methodology, Z.X. (Zhengzheng Xie), J.W., F.G., Z.H. and Z.X. (Zhe Xiang); Supervision, N.Z. and C.Z.; Writing—Original Draft, Z.X. (Zhengzheng Xie), J.W. and F.G.; Writing—Review and Editing, N.Z., Z.H., Z.X. (Zhe Xiang) and C.Z. All authors have read and agreed to the published version of the manuscript.

**Funding:** This research was funded by the National Natural Science Foundation of China (52104104).

**Institutional Review Board Statement:** Not applicable.

**Informed Consent Statement:** Not applicable.

**Data Availability Statement:** The data are available from the corresponding author on reasonable request.

**Acknowledgments:** We would also like to thank the anonymous reviewers for their valuable comments and suggestions that lead to a substantially improved manuscript.

**Conflicts of Interest:** The authors declare that they have no conflict of interest.

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
