# Peer review of "Study on Time-Dependent Failure Mechanisms and CBAG Differential Support Technology of Roadway in Steeply Inclined Coal Seam"

_processes, doi:10.3390/pr11030866_

Round 1

Reviewer 1 Report

[101] The location of the mine is not shown on the small-scale map.

[251-252] Other conditions are not considered. Usually, horizontal stresses exceed vertical ones due to geotectonic processes. Sichuan Province is characterized by a complex geodynamic situation in terms of the development of horizontal stresses in the Earth's crust (Heidbach, O., Rajabi, M., Reiter, K., Ziegler, M. (2016): World Stress Map 2016, GFZ Data Service, doi: 10.5880/WSM.2016.002)

Author Response

Dear reviewer,

Thanks a lot for your valuable comments regarding our submission entitled ‘Study on time-dependent failure mechanism and CBAG differential support technology of roadway in steeply inclined coal seam’. We have seriously considered the suggestions and replied as follows.

Responses to comments:

1. Response to comment: [101] The location of the mine is not shown on the small-scale map.

Thank you for your suggestion, We have checked the small-scale map. Zhaojiaba Coal Mine is Located in Guangyuan City, Sichuan Province, China. Limited to the scale, the mine coincides with the city, and only Guangyuan City is marked.

2. Response to comment: [251-252] Other conditions are not considered. Usually, horizontal stresses exceed vertical ones due to geotectonic processes. Sichuan Province is characterized by a complex geodynamic situation in terms of the development of horizontal stresses in the Earth's crust (Heidbach, O., Rajabi, M., Reiter, K., Ziegler, M. (2016): World Stress Map 2016, GFZ Data Service, doi: 10.5880/WSM.2016.002).

Thank you for your suggestion. I quite agree with you and the article you recommended. However, the horizontal stress measured in Zhaojiaba Coal Mine is very close to the vertical stress, so the pressure measuring coefficient is taken as 1 for convenience of calculation.

Reviewer 2 Report

The selection of support for inclined roadways is particularly important for hard coal deposits due to the frequent stratification of the rock mass. The presented research issues are interesting in terms of science and technology. The valuable parts of the article are the results of numerical research on the extent of the cracks zone and the application of the excavation strengthening method with the use of grouting hollow cable, which significantly contributes to the reduction of excavation deformation. Below are some comments and suggestions:

1. Some information on scale model studies for inclined drifts should be added in the introduction;

2. In the subsection 2.1, it should be specified the distance from the ventilation roadway to other excavations or goafs;

3. In the subsection 2.2, it should be written how the anchors are fixed along the entire length or in sections and with what binder/resin cartridges;

4. In the subsection 2.3, it should be determined by what percentage the cross-section of the roadways changes after deformation;

5. In the subsection 3.1, it should be written what was the value of vertical and horizontal stresses and whether the contact between individual layers was modeled;

6. Figure 6, a full description should be added on the vertical axis;

7. Figures 7, 8 and 9, add the unit to the legend;

8. For the figure, please add information about the extent of cracks around the roadway; c

9. Formula 9, please correct the entry;

10. In the subsection 4.2 it should be written whether the inclination of the layers was taken into account in the analytical considerations;

11. In the subsection 5.1, it should be written types of binder and pump used to fix the hollow grouting cable;

12. In subsection 5.3, a description of sensors and apparatus for measuring the load on anchors should be added;

13. In the discussion subchapter, information should be added that the selection of the roadway support in conditions of high rock mass deformation should consist in common fulfillment of the geometric, ventilation and geomechanical criteria (doi:10.3390/en15103774);

14. One statement should be added to the conclusions regarding the recommendation and the conditions under which new grouting cable support should be used.

Author Response

Dear reviewer,

Thanks a lot for your valuable comments regarding our submission entitled ‘Study on time-dependent failure mechanism and CBAG dif-ferential support technology of roadway in steeply inclined coal seam’. We have seriously considered the suggestions and made revisions as required. In the modified part, we have marked up using the “Track Changes” function, so that changes are easily visible to the reviewer. We do hope our work is in line with your expectations.

Responses to comments:

1. Response to comment: Some information on scale model studies for inclined drifts should be added in the introduction.

Thank you for your suggestion. The relevant content has been added in lines 75-78.

2. Response to comment: In the subsection 2.1, it should be specified the distance from the ventilation roadway to other excavations or goafs.

Thank you for your suggestion. The relevant content has been added in line 98.

3. Response to comment: In the subsection 2.2, it should be written how the anchors are fixed along the entire length or in sections and with what binder/resin cartridges.

Thank you for your suggestion. Related contents have been supplemented and modified in subsection 2.2.

4. Response to comment: In the subsection 2.3, it should be determined by what percentage the cross-section of the roadways changes after deformation.

Thank you for your suggestion. The relevant content has been added in lines 147-148.

5. Response to comment: In the subsection 3.1, it should be written what was the value of vertical and horizontal stresses and whether the contact between individual layers was modeled.

Thank you for your suggestion. The relevant content is in lines 258-263.

6. Response to comment: Figure 6, a full description should be added on the vertical axis.

Thank you for your suggestion. Figure 6 has been modified accordingly in line 276.

7. Response to comment: Figures 7, 8 and 9, add the unit to the legend.

Thank you for your suggestion. Figures 7, 8 and 9 have added the unit to the legend.

8. Response to comment: For the figure 10, please add information about the extent of cracks around the roadway; c.

Thank you for your suggestion. The detailed distribution of cracks in zone A and zone B of roadway can be clearly seen in figure 10a, which can meet the needs of readers.

9. Response to comment: Formula 9, please correct the entry.

Thank you for your suggestion. k stands for constant , , Formula 9 has been corrected.

10. Response to comment: In the subsection 4.2 it should be written whether the inclination of the layers was taken into account in the analytical considerations.

Thank you for your suggestion. In order to improve the strength and stiffness of the anchorage bearing layer and resist disturbance of mining stress and long-term creep effect of soft rock, the bi-directional linkage control of the deep and shallow displacement is realized by increasing the thickness of the foundation anchorage layer and restricting the large displacement in the shallow part with the small displacement in the deep part, which is suitable for a roadway under any surrounding rock inclination angle. The relevant content has been added in lines 430-431.

11. Response to comment: In the subsection 5.1, it should be written types of binder and pump used to fix the hollow grouting cable.

Thank you for your suggestion. The relevant content has been added in lines 476-479.

12. Response to comment: In subsection 5.3, a description of sensors and apparatus for measuring the load on anchors should be added.

Thank you for your suggestion. The relevant content has been added in lines 549-551.

13. Response to comment: In the discussion subchapter, information should be added that the selection of the roadway support in conditions of high rock mass deformation should consist in common fulfillment of the geometric, ventilation and geomechanical criteria (doi:10.3390/en15103774).

Thank you for your suggestion, the following article has been added to the discussion subchapter of the article to enrich the article and improve its quality.

14. Response to comment: One statement should be added to the conclusions regarding the recommendation and the conditions under which new grouting cable support should be used.

Thank you for your suggestion. The relevant content has been added in lines 658-660.
